Suicidal ideation and suicide attempts among university students in South Korea during the COVID-19 pandemic: the application of interpersonal-psychological theory and early maladaptive schema

http://orcid.org/0000-0002-3908-2678 Ha Jeongmin 1
http://orcid.org/0000-0002-8775-0298 Park Dahye 2 dhpark@semyung.ac.kr
1 Department of Nursing, Dong-A University , Busan , South Korea
2 Department of Nursing, Semyung University , Jecheon , South Korea
Khosravi Mohsen
Electronic publication date: 2022 Jul 27
Publication date: 2022
Volume: 10
Electronic Location ID: e13830
Received 2022 May 23; Accepted 2022 Jul 11
Copyright: © 2022 Ha and Park
Copyright year: 2022
Copyright holder: Ha and Park
License: This is an open access article distributed under the terms of the Creative Commons Attribution License, which permits unrestricted use, distribution, reproduction and adaptation in any medium and for any purpose provided that it is properly attributed. For attribution, the original author(s), title, publication source (PeerJ) and either DOI or URL of the article must be cited.
License URL: https://creativecommons.org/licenses/by/4.0/

Keywords: Suicide, COVID-19 pandemic, Interpersonal-psychological theory, Early maladaptive schema

Funding: Ministry of Education of the Republic of Korea National Research Foundation of Korea NRF-2021R1I1A3051439 This work was supported by the Ministry of Education of the Republic of Korea and the National Research Foundation of Korea (NRF-2021R1I1A3051439). The funders had no role in study design, data collection and analysis, decision to publish, or preparation of the manuscript.

==============================
Background

This study examined the application of interpersonal-psychological theory and early maladaptive schema of suicidal ideation and suicide attempts in South Korean university students.

Methods

In this cross-sectional study, data from 367 university students were surveyed using the Interpersonal Needs Questionnaire, Early Maladaptive Schema, Suicide Ideation Scale, and the Acquired Capability for Suicide Scale. Data were collected between June 21 and July 21, 2021.

Results

University students’ interpersonal needs and early maladaptive schema were significantly associated with suicidal ideation, and influencing suicide attempts. The acquired capability for suicide moderated the relationship between suicidal ideation and attempts.

Conclusions

In suicide prevention programs for university students, it is critical to consider their interpersonal needs and early maladaptive schema, and the acquired capability for suicide, to prevent suicidal ideation and attempts among them.

Introduction

Recently, there have been concerns worldwide that the coronavirus disease (COVID-19) pandemic may increase the risk of suicide (Gunnell et al., 2020). Additionally, it has been reported that suicidal ideation is increasing among adults due to the COVID-19 pandemic (Czeisler et al., 2021; Fortgang et al., 2021). Furthermore, a study analyzing suicidal tendencies due to the COVID-19 pandemic emphasized the need to consider its impact on suicidal ideation and self-harm among the youth (John et al., 2020).

South Korea has the highest suicide mortality rate among member countries of the Organization for Economic Co-operation and Development (OECD), at 24.6 per 100,000 people (OECD, 2020). Furthermore, exploring the trend over the past 30 years, the suicide mortality rate in most OECD countries has decreased by 30%; however, it has increased in South Korea (OECD, 2020). On investigating the proportion of deaths due to suicide among the deceased, differentiated by life stage, suicide has been identified as the leading cause of death among Koreans in their 20s since 2007 (Statistics Korea, 2019). This indicates the severity of the suicide problem among youth and university students in South Korea.

Various factors affect suicide among university students, including demographic and economic factors and mental health (Uchida & Uchida, 2017). Interpersonal relationships are one of the primary reasons for suicide. Joiner et al. (2005) explain that suicidal ideation is caused by experiencing a feeling of being burdened from feeling incompetent and feelings of thwarted belongingness due to not being a valuable member of the group. This leads to suicide when the acquired capability for suicide, decreased physical pain tolerance, and fearlessness of death are added to the aforementioned feelings.

Adolescence is a period of forming relationships and developing intimacy within those relationships (Erikson, 1994). However, social distancing due to COVID-19 has reduced university students’ opportunities to have a social life (Kim & Park, 2021). These changes to university students’ daily lives are likely to cause mental health problems.

A qualitative study on COVID-19-related stress among South Korean university students found that they experienced stress and anxiety due to concerns about poor academic quality, social disruption, the decline in employment, and health and safety (Kim & Park, 2021). Additionally, in a survey of 195 university students in the United States, 91% reported stress, anxiety, depression, and fear of losing relationships with loved ones due to the COVID-19 pandemic, and 86% reported a decrease in social interaction (Son et al., 2020). Based on this, the interpersonal-psychological theory of suicide (Joiner et al., 2005; Van Orden et al., 2012) may be a helpful model for explaining suicide among adolescents during the COVID-19 pandemic.

Another interpersonal-related factor affecting suicide is the early maladaptive schema. Early maladaptive schema is an evaluation of oneself and others, a collection of memories, emotions, body sensations, and cognitions related to childhood developmental topics such as abandonment, abuse, neglect, and rejection (Young & Brown, 2005). Early maladaptive schema is a factor related to past interpersonal relationships and affects mental health problems, including depression, anxiety, and even suicidal ideation (Rezaei, Ghazanfari & Rezaee, 2016; Khosravani et al., 2019; Kaya & Aydin, 2021).

Recently, the three-step theory (3ST) of suicide was developed by Klonsky & May (2015). Accordingly, the first step of suicide is the development of suicide ideation, the second step is strong vs. moderate ideation, and the final step is progression from ideation to attempts. Suicide ideation and suicide attempts are strong antecedents of suicide (Klonsky, May & Saffer, 2016). Thus, we should pay attention to suicidal ideation and suicide attempts to explain suicide.

However, few studies have simultaneously identified early maladaptive schema and Joiner’s theory as correlated factors leading to suicidal ideation and attempts.

Objective

This study aims to verify a model in which interpersonal-related factors explain suicidal ideation and attempts among university students during the COVID-19 pandemic, by utilizing early maladaptive schema and Joiner’s interpersonal-psychological theory of suicide.

Materials and Methods

Study design

This cross-sectional survey study aims to construct and verify a model, to explain and predict suicide among university students during the COVID-19 pandemic. This model is developed utilizing early maladaptive schema and Joiner’s interpersonal-psychological theory.

This structural model study verified the moderating effect of acquired capability on the pathway to increasing suicidal ideation and suicide attempts (Fig. 1).

Figure 1 The study’s conceptual framework based on Joiner’s interpersonal needs model.

Study participants

University students aged 19 or older, currently enrolled in a university in South Korea, who understood this study’s purpose, agreed to participate in the study, and were able to respond to the questionnaire, were included in this study using convenience sampling. University students who had difficulties communicating and responding to questionnaires due to severe stress or psychotic symptoms were excluded from the study. For the structural equation model, the appropriate sample size is considered 10–20 times the number of observed variables or 150–400 participants based on the maximum likelihood estimation (Woo, 2014). In this study, 120–240 participants were required for 12 observed variables. We received 367 responses; all 367 were included for data analysis to secure a sufficient sample.

Study measures

Interpersonal needs

The Interpersonal Needs Questionnaire’s Korean version (K-INQ) was used as a self-report scale to evaluate feelings of burden and thwarted belongingness. Van Orden et al. (2012) validated the seven-point scale of 25 items developed by Joiner et al. (2009) to 15 items, and Lee, Lee & Oh (2015) completed the K-INQ with 13 items. In this study, the 13-item K-INQ was used after obtaining permission from the authors. Based on how they were feeling after the COVID-19 pandemic, students were requested to mark each item from one to seven (one indicated “strongly disagree” and seven indicated “strongly agree”) depending on the extent to which they agreed with the statements. In Van Orden et al. (2012) and Lee, Lee & Oh (2015) studies, the reliability of all the items was 0.85 and 0.86, respectively; in this study the reliability was 0.91.

Early maladaptive schema

To measure early maladaptive schema, the Young Short Questionnaire Short Form (YSQ-Short Form) developed by Young (1999) and adapted by Choi & Lee (2018) was used. The YSQ-Short Form questionnaire comprises 15 sub-factors in 75 questions and is rated on a six-point Likert scale (one indicated “strongly disagree” and six indicated “strongly agree”) depending on how well each statement represents the participants’ views. The higher the score, the more the characteristics of the schema measured are reflected. In this study, the reliability was 0.97.

Suicidal ideation and suicide attempt

To measure suicidal ideation, a tool adapted by Kim (2002) from the Suicide Ideation Scale (SIS) developed by Harlow, Newcomb & Bentler (1986) was used after obtaining permission from the author. This is a five-point Likert scale (one indicated “strongly disagree” and five indicated “strongly agree”), wherein higher scores are indicative of stronger suicidal ideation. This scale comprises five questions, including the following, “I have thought about suicide,” “I have thought about dying recently,” “I have told someone that I want to attempt suicide,” and “I have thought that my life will end in suicide,” which indicate suicidal ideation, and “I have attempted suicide,” which indicates a suicide attempt. To distinguish between suicidal ideation and a suicide attempt, the sum of suicidal ideation scores from items one to four were used for suicidal ideation, and the last item on the SIS was used to measure the experience of suicide attempts. Harlow, Newcomb & Bentler (1986) determined the validity using factor analysis but did not verify internal reliability. This tool’s internal reliability was 0.81 in Ha (2017) study and 0.85 in this study.

Acquired capability for suicide

To measure the acquired capability for suicide, a scale developed by Van Orden et al. (2008), which was translated into Korean and validated by Jo (2010), was used after obtaining permission from the author. The scale comprises 20 items on a five-point Likert scale (0: very different from me, 4: very similar to me), where higher scores indicated a higher capability for fatal self-injury or suicide. The internal reliability was 0.85 and 0.80 in Van Orden et al. (2008) and Jo (2010) studies, respectively, and 0.81 in this study.

Data collection

This study was approved by the Institutional Review Board of Semyung University (IRB No. SMU-2021-05-001-01), and data were collected between June 21 and July 21, 2021. First, three to four departments from the university were randomly selected, and permission was obtained from the department heads to collect data through phone calls and visits. Subsequently, after explaining the study’s purpose and data collection method to the professors in charge of the general elective and major subjects, we requested their cooperation. After class, a trained research assistant explained the study’s purpose, content, and methods to students, and only the students who voluntarily agreed to participate in the study were required to sign the written consent and complete the questionnaire. It took approximately 15 min to complete the questionnaire, and a small reward (mobile gift voucher) was provided to the participants.

Data analysis

The data were analyzed using SPSS 24.0 program and AMOS 23.0 (IBM Corp., Armonk, NY, USA). Using the SPSS 24.0 program, the participants’ general characteristics and measurement variables were analyzed using descriptive statistics, and the correlation between variables was analyzed with the Pearson correlation coefficient.

For structural model analysis, AMOS 23.0 was used to identify the goodness of fit for the hypothetical model and the path coefficient’s significance. The analysis is performed based on the Chi-Squared test (χ2), standardized root means square residual (SRMR), normal fit index (NFI), Tucker-Lewis index (TLI), comparative fit index (CFI), goodness-of-fit index (GFI), and adjusted goodness-of-fit index (AGFI) to identify the model’s goodness of fit. NFI, TLI, CFI, GFI, and AGFI, indices of the model’s goodness of fit, show ideal fit if they are 0.9 or higher. If the SRMR is 0.8 or less, it is interpreted as an ideal fit. Multigroup structural equation modeling was used to determine whether the capability for suicide was a moderating variable. A multigroup analysis can help in determining whether the parameter estimates of models differ among groups or whether the relationships established in the model differ according to group affiliation (Kline, 1998).

Results

Suicidal ideation and suicide attempt according to the participants’ general characteristics

The participants’ mean age was 23.38 ± 3.56 years; 21% were male and 79% were female students. The first, second, third, and fourth grades accounted for 14.7%, 20.4%, 23.7%, and 41.1%, respectively, and the middle subjective economic level was the largest at 80.4%. The residence composition was as follows: 83.7% of the participants lived with at least two people, while 16.3% lived alone.

Of the participants, 3% had been infected with COVID-19, 48.8% had been tested for COVID-19, and 11.7% had a self-isolation experience.

There were significant differences in suicidal ideation among the participants based on their gender, grade, economic status, COVID-19 test experience, and self-isolation experience; female students had significantly higher suicidal ideation than male students (t = −2.16, p = 0.031). Additionally, it was found that the third grade had higher suicidal ideation than the first grade (F = 3.50, p = 0.016), the lower the economic status, the higher the suicidal ideation (F = 3.38, p = 0.035), and the more self-isolation experience due to COVID-19, the higher the suicidal ideation (t = 2.21, p = 0.028).

There was a significant difference in suicide attempts based on residence composition, and it was found that suicide attempts were significantly higher in those living alone than in those living with at least two people (t = −65, p = 0.009) (Table 1).

Table 1 Suicidal ideation and attempt according to the participants’ general characteristics.

Characteristics	Categories	n (%) or
M ± SD	Suicidal ideation	Suicide attempt	
M ± SD	t or F (p)	M ± SD	t or F (p)	
Age (year)		23.38 ± 3.56	7.90 ± 3.58	1.375 (0.136)	1.51 ± 0.88	1.128 (0.321)	
Gender	Male
Female	77 (21.0)
290 (79.0)	7.11 ± 3.15
8.10 ± 3.66	−2.161 (0.031)*	1.49 ± 0.09
1.52 ± 0.05	−0.209 (0.834)	
Grade	1st
2nd
3rd
4th	54 (14.7)
75 (20.4)
87 (23.7)
151 (41.1)	6.63 ± 3.46
7.60 ± 3.68
8.44 ± 3.61
8.18 ± 3.46	3.500 (0.016)*
3 > 1	1.26 ± 0.76
1.55 ± 0.87
1.63 ± 0.94
1.52 ± 0.88	2.067 (0.104)	
Residence	1
2
3
4
5
6
7
8	113 (30.8)
37 (10.1)
5 (1.4)
15 (4.1)
115 (31.3)
49 (13.4)
9 (2.5)
24 (6.5)	7.76 ± 3.35
7.16 ± 3.37
8.40 ± 4.16
7.67 ± 2.50
8.09 ± 3.57
8.45 ± 4.33
7.56 ± 4.30
7.79 ± 3.73	0.492 (0.841)	1.49 ± 0.86
1.35 ± 0.59
1.60 ± 0.89
1.20 ± 0.56
1.58 ± 0.90
1.63 ± 1.20
1.56 ± 0.88
1.45 ± 0.72	0.710 (0.664)	
Economic status	Low
Middle
High	54 (14.7)
295 (80.4)
18 (4.9)	8.81 ± 3.81
7.82 ± 3.55
6.44 ± 2.66	3.375 (0.035)*	1.56 ± 0.95
1.52 ± 0.89
1.28 ± 0.57	0.705 (0.495)	
Family composition	Living alone
Living with at least two people	60 (16.3)
307 (83.7)	8.03 ± 3.33
7.87 ± 3.63	0.324 (0.747)	1.78 ± 1.01
1.45 ± 0.84	2.649 (0.009)*	
COVID-19 confirmed experience	Yes
No	3 (0.8)
364 (99.2)	11.33 ± 5.03
7.86 ± 3.56	1.674 (0.095)	2.00 ± 1.00
1.50 ± 0.88	0.960 (0.338)	
COVID-19 test experience	Yes
No	179 (48.8)
188 (51.2)	8.27 ± 3.81
8.74 ± 3.32	1.978 (0.049)*	1.53 ± 0.91
1.50 ± 0.86	0.272 (0.786)	
Self-isolation experience	Yes
No	43 (11.7)
324 (88.3)	9.02 ± 4.00
7.74 ± 3.50	2.209 (0.028)*	1.70 ± 0.94
1.49 ± 0.87	1.467 (0.143)	
Note:

An asterisk (*) indicates p < 0.05.

Correlation between perceived burdensomeness, thwarted belongingness, early maladaptive schema, acquired capability, suicidal ideation, and suicide attempts

Suicidal ideation had a significant positive correlation with perceived burdensomeness (r = 0.64, p < 0.001), thwarted belongingness (r = 0.36, p = 0.001), and acquired capability (r = 0.28, p < 0.001). Suicide attempts were significantly positively correlated with perceived burdensomeness (r = 0.37, p < 0.001), thwarted belongingness (r = 0.17, p = 0.001), acquired capability (r = 0.19, p < 0.001), and suicidal ideation (r = 0.55, p < 0.001). Perceived burdensomeness showed a significant positive correlation with thwarted belongingness (r = 0.59, p < 0.001) and acquired capability (r = 0.22, p < 0.001), but there was no significant correlation with early maladaptive schema (r = −0.00, p = 0.940) (Table 2).

Table 2 Correlations among the major variables.

Variables	Perceived burdensomeness	Thwarted belongingness	Early maladaptive schemas	Acquired capability for suicide	Suicidal ideation	Suicide attempts	
Perceived burdensomeness	1						
Thwarted belongingness	0.59 (p < 0.001)	1					
Early maladaptive schemas	−0.00 (0.940)	0.05 (0.940)	1				
Acquired capability	0.22 (p < 0.001)	−0.01 (0.840)	−0.06 (0.246)	1			
Suicidal ideation	0.61 (p < 0.001)	0.36 (p < 0.001)	−0.06 (0.246)	0.28 (p < 0.001)	1		
Suicide attempts	0.37 (p < 0.001)	0.17 (p = 0.001)	−0.31 (0.056)	0.19 (p < 0.001)	0.55 (p < 0.001)	1	

Structural equation modeling

Pathway analysis between interpersonal needs, suicidal ideation, and suicide attempts

As a result of analyzing the pathway of interpersonal needs, early maladaptive schema, suicidal ideation, and suicide attempts, based on this study’s model, the model’s goodness of fit was χ2 = 2.915 (p < 0.001), SRMR: 0.007, GFI: 1.000, AGFI: 0.998, TLI: 1.020, NFI: 0.999, and CFI: 1.000, which satisfied the fit criteria of the model (Table 3, Fig. 2).

Table 3 Fitness of the models testing theory of suicide.

	χ2	p	Standardized root mean square residual	Goodness-of-fit index	Adjusted
goodness-of-fit index	Tucker–Lewis index	Normed fit index	Comparative fit index	
Research model	2.915	0.000	0.007	1.000	0.998	1.020	0.999	1.000	
	16.837	0.010	0.068	0.985	0.948	0.946	0.967	0.978	

Figure 2 Verification of research model.

Moderating effect of acquired capability for suicide on the relationship between suicidal ideation and suicide attempts

To verify the moderating effect, it was classified into low and high groups based on the mean value of the acquired capability for suicide, which was a moderating variable. The results of the equal constrained model, in which the path coefficients of the two groups were equally constrained, were χ2 = 16.837 (p = 0.010), SRMR: 0.068, GFI: 0.985, AGFI: 0.948, TLI: 0.946, NFI: 0.967, and CFI: 0.978, showing the suitability of the relationship presented in this study’s conceptual framework (Table 3, Fig. 1).

Analysis of the difference in the path coefficients between the low and high groups with acquired capability for suicide

The analysis revealed a significant difference between the two pathways in the low and high groups. The extent of change in χ2 between the equal constrained and non-constrained models were df = 1 to 7.057, indicating a significant moderating effect. The higher the suicidal ideation, the more the number of suicide attempts was not significant in the group with low acquired capability for suicide, but significant in the group with high acquired capability for suicide. Therefore, the acquired capability for suicide significantly regulates the relationship between suicidal ideation and suicide attempts (Table 4).

Table 4 Difference in path coefficients between low and high acquired capability for suicide groups.

Path	df	Δ χ2	Coefficient	
Low group (β1)	High group (β2)	
Suicidal ideation →
Suicide attempt	1	7.057	0011	12.169 (<0.001)*	
Note:

An asterisk (*) indicates p < 0.001.

Discussion

This cross-sectional survey study aimed to construct and verify a model to explain and predict suicide among university students during the COVID-19 pandemic by utilizing early maladaptive schema and Joiner et al. (2005) interpersonal-psychological theory of suicide. The results showed that interpersonal needs and early maladaptive schema influenced suicidal ideation, and that the acquired capability for suicide was a factor in controlling the relationship between suicidal ideation and suicide attempts. Overall, it was found that the hypothetical model utilizing both, the early maladaptive schema and interpersonal-psychological theory of suicide was the most suitable for explaining suicide attempts in Korean university students. This study is consistent with a meta-analysis of studies on the relationship between early maladaptive schema and suicidal ideation, identifying that early maladaptive schema is related to suicidal ideation (Pilkington, Younan & Bishop, 2021).

Additionally, this study determined that university students’ experience of living alone and self-isolation had a significant relationship with suicide-related variables. Furthermore, it was found that thwarted belongingness significantly affected suicidal ideation. Consistent with our study, a study of 500 adults in the United States identified that they experienced thwarted belongingness when quarantined at home, which consequently acted as a factor in influencing suicidal ideation (Gratz et al., 2020). These results support the existing theory that frustration with belongingness can increase the desire for suicide and the risk of suicide (Van Orden et al., 2010).

In Gratz et al. (2020) study, the feeling of burdensomeness after the pandemic was not significantly correlated with suicidal ideation; however, this study identified that the feeling of burdensomeness had a significant effect on suicidal ideation, showing a slight difference. This study also demonstrated that a poor economic situation was related to suicidal ideation among university students. This may be due to future-related stress and their economic situation. It was found that suicidal ideation was significantly higher in the third grade than in the first grade. Moreover, suicidal ideation increased significantly until the third grade but decreased slightly in the fourth grade. This seems to be consistent with Hong & Lee (2020) qualitative study results; although South Korean nursing students are under a significant amount of pressure to find employment as they get closer to their graduation, they learn to cope with stress while maturing through various experiences during the fourth year.

In South Korea, the number of university students experiencing stress due to financial reasons has increased since the pandemic because they were unable to find employment (Kim & Park, 2021). University students may have judged themselves as incompetent and considered themselves a burden under these circumstances. Additionally, Classen & Dunn (2012) study demonstrated that although short-term unemployment is not related to an increase in suicide risk, long-term unemployment is related to an increase in suicide risk. However, as this study collected data for a considerable period after the COVID-19 outbreak, it can be interpreted that the results were slightly different due to identifying the long-term effects when compared to Gratz et al. (2020) study.

This study’s results emphasize the relationship between interpersonal needs and early maladaptive schema to the risk of COVID-19–related suicide among South Korean university students. These findings suggest that interpersonal needs and early maladaptive schema may be critical for suicide prevention and intervention. Unemployed university students whose interpersonal relationships were disrupted during COVID-19 and who reported early maladaptive schema should be targeted for intensive suicide prevention and intervention. Additionally, interventions should be implemented to alleviate the negative psychological consequences of social isolation and economic problems that may arise or worsen due to quarantine measures. For example, universities should develop systems (e.g., entrance ceremony using metaverse) through which students can communicate online and thereby experience a sense of belonging (Kim & Park, 2021).

Universities should provide active livelihood support for their students by creating jobs related to their education, such as creating jobs through practical institutions with industry-academic linkage (Kim & Park, 2021). It is also necessary to allow university students to express their emotions in various ways to help them with their uncertainty about the future and employment-related stress, and prepare various systematic department adaptation or employment preparation programs. Furthermore, as childhood trauma and adversity may be related to their interpersonal needs, it is necessary to periodically examine their interpersonal needs or early maladaptive schema to provide an effective intervention with insight into their suicidal ideation (Pilkington, Younan & Bishop, 2021).

Finally, in individuals with high acquired capability for suicide, suicidal ideation directly affected suicide attempts; however, the association was not significant in individuals with low acquired capability for suicide. It is consistent with the interpersonal-psychological theory of suicide (Joiner et al., 2005; Van Orden et al., 2010), which holds that the acquired capability for suicide plays a moderating role in the relationship between suicidal ideation and suicide attempts.

Based on these results, our capability to better predict and prevent suicide depends on a better understanding of the transition from suicidal ideation to attempt. It was also found that intense suicidal ideation leads to suicide only if there are means and capabilities (dispositional, acquired, and practical) to attempt it (Dhingra, Klonsky & Tapola, 2019). For suicide prevention and intervention, the acquired capability level for suicide should be evaluated by assessing a participant’s history of attempted suicide (Bostwick et al., 2016) and should be accompanied by reduced access to lethal means of suicide, such as guns (Anestis & Capron, 2018). As there are more suicides with briquettes and coal firelighters than suicides with guns in South Korea, it is necessary to restrict the distribution and sale of briquettes and coal firelighters, and manage them safely and systematically (Roh & Kang, 2019). It is also essential to train university students to deal with negative emotions in an adaptive way and not resort to suicide (Hu et al., 2019; Kang et al., 2019).

However, considering that in this study suicidal ideation among university students living alone is significantly higher than among those living with their families, it may be challenging to use the aforementioned methods to prevent suicide among those who live alone. Therefore, in a country with a high suicide rate like South Korea, all citizens should receive gatekeeper-related education and actively play their role to effectively prevent suicide. It is also vital to continue funding and deploying staff to crisis hotlines so that individuals with limited social contact can ask for help in emergencies (Gratz et al., 2020). Furthermore, as suicidal ideation was significantly higher among university students with self-isolation experience than those without self-isolation experience in this study, vulnerable individuals should be able to use and access evidence-based remote mental health services during stay-at-home orders and other social distancing adjustments (Reger, Stanley & Joiner, 2020).

Despite the significance of this study, there are several limitations. First, as the participants were all university students living in South Korea, one should be cautious about generalizing the results to other populations. Moreover, most of the participants in this study were women, and all of them were nursing students. A report stated that men attempt suicide in a more lethal way than women (Jordan & McNiel, 2020) and a previous study indicated that there is a difference in the socio-psychological problems experienced by college students in a pandemic situation according to their major (Ha & Park, 2021). It can be inferred that the sample bias makes it difficult to generalize the study results. Therefore, further studies should expand the sampling range to other countries and races. Second, this study cannot infer a causal relationship between variables as a cross-sectional design was used. Additionally, there are limitations to self-report questionnaire data that can be affected by social desirability bias or recall difficulties. Future studies should adopt a clinical interview and longitudinal approach. Third, this study evaluated suicide attempts using a single item of the SIS. Future studies need to further clarify suicidal ideation and attempts by adding a scale for suicide attempts. This study is meaningful in that it theoretically verified the interpersonal psychological theory, including early maladaptive schema for suicide among Korean university students in a pandemic situation. The results can help provide a theoretical rationale for understanding and preventing suicide in university students.

Conclusions

This study provides a theoretical rationale for understanding and preventing suicide among Korean university students during a pandemic. When developing a suicide prevention program for university students, their interpersonal needs and early maladaptive schema should be considered. Additionally, as acquired capability for suicide was a moderating factor in the relationship between suicidal ideation and suicide attempts, the acquired capability for suicide in university students should also be managed in suicide prevention programs.

Supplemental Information

Supplemental Information 1 Raw data.

Click here for additional data file.

Supplemental Information 2 Raw data & coding book.

Click here for additional data file.

The authors are very grateful to the participants who consented to participate in this study.

Additional Information and Declarations

Competing Interests

Author Contributions

Human Ethics

Data Availability

The authors declare that they have no competing interests.

Jeongmin Ha conceived and designed the experiments, performed the experiments, prepared figures and/or tables, authored or reviewed drafts of the article, and approved the final draft.

Dahye Park conceived and designed the experiments, performed the experiments, analyzed the data, prepared figures and/or tables, authored or reviewed drafts of the article, and approved the final draft.

The following information was supplied relating to ethical approvals (i.e., approving body and any reference numbers):

This study was approved by the Institutional Review Board of Semyung University (IRB No. SMU-2021-05-001-01).

The following information was supplied regarding data availability:

The raw measurements are available in the Supplemental Files.

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
