# Peer review of "Suicidal ideation and suicide attempts among university students in South Korea during the COVID-19 pandemic: the application of interpersonal-psychological theory and early maladaptive schema"

_PeerJ, doi:10.7717/peerj.13830_

## Round 0.1 · original submission · Major Revisions

I have now received the reviewers' comments. They suggest some major revisions to your manuscript. Therefore, I invite you to respond to the reviewers' comments and revise your manuscript. Once again, thank you for submitting your manuscript to PeerJ and I look forward to receiving your revision as soon as possible.

Reviewer 1 ·

Basic reporting

The English used in this research paper is clear and easy to understand. Papers are prepared in polite and technically correct language. The author presents an introductory description that encourages the emergence of research questions. But the basic theory that explains the research concept is not stated. This paper presents an interesting point and seeks to highlight the internal factors that drive suicide in students in South Korea by providing relevant supporting research data. The research data table is clearly presented with an explanatory label, but the presentation does not use an open table. Raw data is shared as a requirement document. The results of the research are shown in the report but without starting with the submission of the hypothesis.

Experimental design

The research conducted is original and relevant to the stated objectives. The research description has explained the gaps from other studies that led to the emergence of this research. The research proposal has passed the ethical review set by the university. The research method was delivered well according to the specified sample size range. The data presented can be explained based on a strong and acceptable scientific field. The various assessment tools used in this study are well explained and easy to understand. The data processing process is presented systematically with a simple delivery.

Validity of the findings

The research is not a replication study. The results of the study produce new findings that can be used as a basis for further research by other researchers. Conclusions are conveyed properly and firmly with conformity that answers the research objectives.

Additional comments

Suggestions for improvement are that researchers can improve the way the data is presented with open table images and add an explanation of the basic theory regarding the contribution of factors behind suicidal behavior in young adults or college students.

Reviewer 2 ·

Basic reporting

I would like to thank you for the opportunity to review this manuscript. The present study aimed at examined the application of interpersonal-psychological theory and early maladaptive schema of suicide in South Korean university students.
I read the main text of the article but i did not find any innovation in the results. I believe that this paper is not a significant contribution that explains these findings more than previous research has already done. So, I do agree the importance of this topic, but I'm not sure how your added research will accomplish this since the data was already in existence and you did not contribute significantly to adding more information to better manage this difficult situation.

Experimental design

There are also some fundamental methodological flaws in the design, execution, and the analysis of the results. For example, it is unclear whether the study is powered to conduct the number of analyses undertaken. Could the authors provide a power analysis or justify why they conducted their analyses without sufficient power? Calculating the sample size by power analyzing (see Faul et al. 2009, PMID: 19897823) could certainly increase the accuracy of the study.

Validity of the findings

Regardless of the results obtained from the model, it is crucial to emphasize that accurate predictions cannot be guaranteed by cross-sectional study. Rather, development of prediction models is based on cohort study. Thus, prediction models resulting from cross-sectional designs can be misleading.

Additional comments

The format and structure of the manuscript should be improved as is mentioned in the author guideline for manuscript submission.
Finally, I suggest that the manuscript is proofread by a fluent English speaker or Editing service. So please make sure there are no English errors.

·

Basic reporting

The paper is written in a straightforward manner and is readily understood. The language of the article is appropriate for the subject matter, particularly as the issue of suicide and suicidal ideation may provoke significant distress in some readers.

The references are appropriate and considered. I note the authors have referenced articles that raise concern regarding the risk of an increase in suicide rates during the pandemic.

Areas to address in the paper.

It would be helpful if the authors considered publications that have not found an overall increase but have noted variations related to social and gender issues. Pirkis et al. have published a number of papers in relation to suicide data during the pandemic, with the papers referring to a large number of jurisdictions.

Experimental design

The objective of the study is identified as evaluating the relationship between interpersonal factors and maladaptive schema/Joiner’s theory and suicide.

A point of concern is that, in reality, the study seems to be addressing the issue of suicidal ideation/attempts and its relationship to the measures used in the study. My understanding of the paper is that it is cross-sectional; therefore, it is unknown if any of the students died by suicide; rather, we know if they have experienced ideations or self-harm attempts.

It would be helpful if the authors reviewed this aspect.

However, the study addresses important issues concerning factors relevant to suicidal ideation/attempt and, therefore, the risk of suicide in an at-risk population.

The study design is appropriately explained, as are the data analysis methods.

Validity of the findings

Based on the data and the analysis, the findings are significant and add to our knowledge of risk factors for suicidal ideation/attempts.

Areas to address in the paper.

I note that the students came from a number of different years and groups. While the data related to the years were analysed, it would be helpful to know whether there were differences between the student groups, particularly as line 242 refers specifically to nursing students. Does their data differ from the other student groups?

Of concern in the study is the bias towards females, i.e. they are overrepresented 79% vs 21% (males). This aspect is relevant and should be part of the discussion as female self-harm rates are much greater than male self-harm rates in many countries ( I am unsure if this is different in South Korea), but for most countries, males account for three-quarters of all deaths by suicide. Do the findings of this study address these aspects?

Additional comments

Areas to address in the paper.

My main comments centre on the issue of the impact of the gender bias in the data, which does not appear to be adequately explored, and that the objectives outlined in the paper indicate it is about the relationship between suicide and a number of measured factors assessed by self-report questionnaires. I would suggest that the authors consider discussing it as the relationship between suicidal ideation/attempts, which are risk factors for suicide.

It would be helpful and expand on the paper if the authors provided information regarding any change in suicide rates in South Korea across all age groups and, similarly, if the rates of self-harm presentation to hospitals have varied since COVID-19 impacted South Korea. The impact of the study could be broadened by comparison with other countries apart from the USA.

The authors have succinctly documented the strengths and limitations of the study; however, the gender issues should be included.

I want to thank the authors for the opportunity to review their paper. Their findings are of considerable interest and add to our knowledge of risk factors.

I would be pleased to re-review this paper and consider the authors’ responses.

---

## Round 0.2 · accepted · Accept

The comments given have been looked into and the paper has been revised. Thank you.

Reviewer 2 ·

Basic reporting

no comment

Experimental design

no comment

Validity of the findings

no comment

Additional comments

In my opinion this manuscript has been revised with attention to the reviewers' comments and can now be published.